# Value and Diagnostic Efficacy of Fetal Morphology Assessment Using Ultrasound in a Poor-Resource Setting

**DOI:** 10.3390/diagnostics9030109

**Published:** 2019-09-01

**Authors:** Ofonime N. Ukweh, Theophilus I. Ugbem, Chibuike M. Okeke, Ernest U. Ekpo

**Affiliations:** 1Department of Radiology, College of Medical Sciences, University of Calabar, Calabar 540242, Nigeria; 2Department of Histopathology, College of Medical Sciences, University of Calabar, Calabar 540242, Nigeria; 3Department of Family Medicine, College of Medical Sciences, University of Calabar, Calabar 540242, Nigeria; 4Faculty of Health Sciences, The University of Sydney, Discipline of Medical Radiation Sciences, Cumberland Campus, 75 East Street, Lidcombe NSW 2141, Australia; 5Orange Medical Diagnostics, Clinical Laboratory and Research Center, Marian Road, Calabar 540222, Nigeria

**Keywords:** congenital anomalies, prenatal diagnosis, morphology survey, morphology ultrasound

## Abstract

Background: Ultrasound is operator-dependent, and its value and efficacy in fetal morphology assessment in a low-resource setting is poorly understood. We assessed the value and efficacy of fetal morphology ultrasound assessment in a Nigerian setting. Materials and Methods: We surveyed fetal morphology ultrasound performed across five facilities and followed-up each fetus to ascertain the outcome. Fetuses were surveyed in the second trimester (18th–22nd weeks) using the International Society of Ultrasound in Obstetrics and Gynecology (ISUOG) guideline. Clinical and surgical reports were used as references to assess the diagnostic efficacy of ultrasound in livebirths, and autopsy reports to confirm anomalies in terminated pregnancies, spontaneous abortions, intrauterine fetal deaths, and still births. We calculated sensitivity, specificity, positive and negative predictive values, Area under the curve (AUC), Youden index, likelihood ratios, and post-test probabilities. Results: In total, 6520 fetuses of women aged 15–46 years (mean = 31.7 years) were surveyed. The overall sensitivity, specificity, and AUC were 77.1 (95% CI: 68–84.6), 99.5 (95% CI: 99.3–99.7), and 88.3 (95% CI: 83.7–92.2), respectively. Other performance metrics were: positive predictive value, 72.4 (95% CI: 64.7–79.0), negative predictive value, 99.6 (95% CI: 99.5–99.7), and Youden index (77.1%). Abnormality prevalence was 1.67% (95% CI: 1.37–2.01), and the positive and negative likelihood ratios were 254 (95% CI: 107.7–221.4) and 0.23 (95% CI: 0.16–0.33), respectively. The post-test probability for positive test was 72% (95% CI: 65–79). Conclusion: Fetal morphology assessment is valuable in a poor economics setting, however, the variation in the diagnostic efficacy across facilities and the limitations associated with the detection of circulatory system anomalies need to be addressed.

## 1. Introduction

Congenital anomalies are a significant cause of deaths [1,2]. According to the World Health Organisation, congenital anomalies account for over 17–42% of deaths within the first month of life across the world [2]. The literature shows that 2.5% of deaths in neonates within seven days of life result from congenital anomalies [3]. For survivors, birth defects lead to significant physical, psychological, emotional, and social impairment [4], which have a negative impact on the quality of life [5,6]. Congenital anomalies vary across ethnicities [7,8], however, there is limited data on their prevalence in resource-poor countries. Risk factors for congenital anomalies include genetic signatures [7], age at pregnancy, obesity, smoking, and nutrition, as well as metabolic conditions [9,10].

Advances in medicine have improved early detection and management of fetuses with congenital anomalies [11,12]. Early detection and management of these conditions have been shown to improve fetal outcomes, leading to recommendations for prenatal screening in many countries [13]. Ultrasound is the frontline modality for early detection of congenital anomalies due to its high sensitivity, non-ionizing nature, cost-effectiveness and availability [14]. This modality, when utilized in the second trimester facilitates the detection and follow-up of a wide array of anomalies, whilst simultaneously assessing other obstetric parameters such as amniotic fluid volume, number of fetuses and fetal position [14]. Despite the important advantages of ultrasound in prenatal screening, its efficacy depends on the skills and expertise of the imaging personnel and varies between personnel [15]. When nested within operators, previous studies have shown that the sensitivity of ultrasound varies according to body parts, being highest for central nervous and musculoskeletal systems with >90% sensitivity, and lowest for circulatory system malformations, where it is less than 50% [15,16,17].

There is considerable variation in obstetric care in Nigeria due to differences in socio-cultural, economic, and geographical distribution of the population [18,19]. Women in rural areas access antenatal services provided by community health centers, which often do not have the facilities and expertise to undertake detailed obstetric assessment and care [18,19]. Access to health antenatal services is further challenged by the poor economic status and deep-rooted socio-cultural beliefs [19,20]. With the Mother and Child Initiative of the General Electric Healthcare, where community nurses are trained to detect risk in pregnancy using ultrasound, obstetric care is expected to improve. Conversely, women in suburban and urban regions have access to public and private antenatal services and have been shown to demonstrate good uptake of obstetric ultrasound assessment [21,22]. Studies have also shown improvement obstetric outcomes for women in these regions [23], however, progress has been slow, partly due to the absence of government-funded programs to improve obstetric care.

Generally, early detection approaches are not well developed in low resource settings such as Africa due to a limited access to screening tools and expertise to inform diagnosis [24,25]. Due to a paucity of radiologists, most ultrasound investigations are performed by sonographers and resident radiologists, and some of these personnel have limited training or experience in ultrasonography [24,25]. A previous work has also reported that ultrasound training in middle and low-resource countries falls short of the criteria recommended by the World Health Organisation in terms of mentorship, duration of training, and quality control [25]. In addition, most of the available ultrasound facilities have limited resources to perform advanced ultrasound assessments [24]. Whilst early detection can improve management and outcome of congenital anomalies, we need to ensure that early detection strategies are valuable and there is expertise to inform diagnosis and guide clinical decision-making.

In developing countries such as those in Africa, there are no tailored fetal morphology ultrasound screening programs. However, some practices perform fetal morphology assessment as part of a routine obstetric evaluation or upon request. The limited ultrasound expertise and lack of quality control in middle and low-resource settings emphasise the need to investigate the diagnostic accuracy of practices undertaking fetal morphology assessment and the value of implementing such a screening program in such settings. However, to our knowledge, there is no published data on the diagnostic accuracy of ultrasound practitioners in Sub-Saharan Africa in fetal morphology evaluation and the value of such assessments in this setting is unclear. We need to address these gaps so that interventions to transform prenatal fetal morphology assessment and management can be sought. This work aims to assess the diagnostic performance of facilities undertaking fetal morphology survey and the value of implementing a fetal morphology screening program in a Nigerian setting.

## 2. Materials and Methods

A prospective assessment of performance of diagnostic facilities in detecting fetal anomalies using ultrasound was conducted from January, 2016 to February, 2019. We recruited pregnant women who were in their second trimester (18th to the 22nd week of gestation) and consented to the study. The recruitment of women in this stage of gestation is based on the International Society of Ultrasound in Obstetrics and Gynecology (ISUOG) recommendation for fetal morphology survey at this gestational stage [26]. For each of these participants, we recorded maternal age, gestational age, and number of pregnancies. We did not collect other clinical information since the aim of our study was not to establish the risk factors for congenital anomalies.

Ultrasound examinations were performed in the dorsal decubitus position, but with patients changing orientation as directed by sonographers or radiologists to facilitate better evaluation of fetal morphology. Morphological surveys of all fetuses were performed between the 18th to the 22nd week of pregnancy according to the ISUOG guideline [26]. For cranial anomalies, the ultrasound practitioners assessed the size, shape, integrity and density of the cranial vault. Other structures assessed included brain tissues such as the cerebrum, cerebellum, cavum septi pellucidi, ventricles, thalami, falx cerebri, and cistern magna. Facial anomalies assessment included evaluation of the upper lip, nasal profile and nose. Assessors also evaluated the neck for masses. Spinal, thoracic (thoracic cage and cardiovascular structures), abdominal (kidneys, liver, bladder, gut, gall bladder and cord insertion), limb and genital assessments were performed following the ISUOG guideline. Of the five facilities surveyed, four were operated by sonographers (two in one facility, and one each in the other three facilities) at different stages of practice and the other by senior resident radiologists (*n* = 3). All assessors were familiar with the ISUOG guideline. Each personnel performed the evaluation independently and reported their findings, as this is the practice in the setting investigated. Each of the assessors had performed a minimum of 1000 ultrasound examinations of fetuses at 20 ± 2 weeks of gestation per year within the last three years prior to the current study. Because our primary endpoint was to assess the value of fetal morphology scan and overall accuracy of such assessment in the facilities surveyed as opposed to the accuracy of each ultrasound personnel, each fetus was evaluated by only one personnel. For multiple gestations, a detailed morphological evaluation of each fetus was performed. We then recorded the report for each fetus and followed it up to delivery. Ultrasound examinations were performed using General Electric Versana Essential (General Electric Healthcare, Chicago, IL USA) in two facilities and the other three facilities utilized Mindray scanners: DP-110 Plus, DP 3300, and DP-5 (Shenzhen Mindray Bio-Medical Electronics Co., Ltd., Shenzhen, China). A 3.5 MHz convex transducer was used in all these facilities. Two of these five centers are specialist hospitals and provide ultrasound services to their patients and clients referred from other health facilities. The other three centers are private diagnostic facilities and providing diagnostic imaging and laboratory services to patients from public and private health facilities.

Postnatal findings were used as gold standard for assessing the diagnostic efficacy of morphological ultrasound in these facilities. Normal fetuses were confirmed to be normal using the reports of pediatricians. For livebirths with anomalies, postnatal clinical and surgical reports and supplementary imaging findings were used to confirm internal anomalies detected on ultrasound. Clinical reports of pediatricians were used as a reference to establish the diagnostic efficacy of ultrasound for external fetal anomalies. Autopsy was performed to confirm findings in cases where there was therapeutic termination of the pregnancy, spontaneous abortions, and intrauterine fetal deaths and still births. The autopsy included gross photographs of the fetus, whole body radiography, and external and internal autopsy examinations. Institutional Review Board of Orange Medical Diagnostics approved the study (IRB: 015/OMD/010; Approval date: 08/01/2015).

### Statistical Analysis

Using the clinical, surgical, and autopsy reports as reference standards, we calculated the sensitivity (correct diagnosis of anomalies on ultrasound), specificity (correct report of fetuses without anomalies), and the Area Under the Receiver Operating Characteristic Curve (ROC) to assess the diagnostic efficacy of morphological ultrasound assessments performed at these facilities. ROC AUC was interpreted as follows: 0.90–1 (excellent), 0.80–0.90 (good), 0.70–0.80 (fair), 0.60–0.70 (poor), and 0.50–0.60 (fail).

Since the AUC value obtained from ROC analysis is influenced by sensitivity and specificity, we computed the Youden index to quantify the overall diagnostic performance of all morphological ultrasound assessments. The Youden index was calculated to account for any effect sensitivity and specificity may have on accuracy as described by Unal et al. [27]. We also calculated the positive predictive value (PPV) and negative predictive value (NPV) of ultrasound for congenital anomalies. We also calculated the positive and negative likelihood ratios and post-test probabilities to assess the value of performing morphological assessment in this region. Likelihood ratios were interpreted as described by McGee [28]. To assess the anatomical regions presenting diagnostic difficulties, we calculated the percentage misses and for the systems in which anomalies were detected.

## 3. Results

A total of 6520 fetuses were surveyed for anatomical congenital anomalies, and sample sizes across sites ranged from 599 to 2777 (mean = 1304). These were from women aged between 15–49 years (mean = 31.7 years). A total of 4375 of the women surveyed had at least one child (range 1–4 children) prior to the current study. Years of experience of the ultrasound personnel varied from 3 to 15 years. Of the total number of fetuses surveyed, anomalies were detected in 84 fetuses on ultrasound, and a further 25 additional anomalies were detected postnatally. The anomalies were fairly distributed equally across all ages of women examined. The types and numbers of anomalies in our cohort are shown in Figure 1, and the distribution of anomalies detected on ultrasound and postnatal findings according to body systems is shown in Table 1.

Overall, the prevalence of anomalies in our cohort was 1.67% (*n* = 109/6520 fetuses), and a majority (35%) of these anomalies were related to the central nervous system followed by digestive (12.8%) and genitourinary (11.9%) systems. Examples of the ultrasound features of these anomalies are shown in Figure 2A,B and Figure 3.

Abnormality prevalence across the different sites surveyed varied from 1.6%; 95% CI: 0.85–2.9 (center 1) to 2.2%; 95% CI: 1.4–3.3 (center 3). The mean diagnostic efficacy of morphological ultrasound assessment from all facilities including their 95% confidence intervals is shown in Table 2. The mean positive predictive value was 72.4 (95% CI: 64.7–79.0), the negative predictive value was 99.6 (95% CI: 99.5–99.7), and the Youden index was 77.1%. The positive and negative likelihood ratios were 254.4 (95% CI: 107.7–221.4) and 0.23 (95% CI: 0.16–0.33), respectively.

Table 3 shows the diagnostic performance metrics in all five centers surveyed. Whilst specificity and negative predictive values were almost similar in all centers, we found differences in other diagnostic performance metric across sites: Sensitivity varied from 58 (95% CI: 28.0–85.0) to 83 (95% CI: 63–95), specificity ranged from 99.5 (95% CI: 99–99.8) to 99.7 (95% CI: 99–100), and the Area Under the Curve varied from 78.9 (95% CI: 63.5–92.5) to 91.3 (95% CI: 81.3–97.4). The ranges of other performance metrics across sites are also shown in Table 3.

The likelihood ratios were calculated to account for the effect of disease prevalence on sensitivity and specificity and to establish the post-test probability of detecting fetal anomaly. The post-test probability represents the revised probability of detecting an anomaly after taking into account new gestations [29]. The analysis yielded a post-test probability for a positive test of 72% (95% CI: 65–79), indicating that 1 in 1.4 of fetuses diagnosed with an anomaly on ultrasound does in fact have a congenital malformation, and that fetal morphology ultrasound assessment is valuable.

## 4. Discussion

Advances in treatment of fetal anomalies in-utero and the need for informed decision-making regarding gestations with significant fetal malformations have increased the use of ultrasound for fetal morphological survey. However, it is an open question whether fetal morphology assessment is valuable across all populations given the ethnic differences in risk of fetal anomalies. Also, expertise and technology differ across countries, and it is unlikely that fetal anomalies will be perceived or correctly reported by all ultrasound operators [30], particularly in countries where there is no low expertise. We performed a multi-site analysis to determine if fetal morphology evaluation is valuable in a low-resource setting and if there is expertise to undertake such assessments. Our results show that regardless of the abnormality prevalence, these facilities recorded very high specificity scores, however, there were considerable variations in performance metric such as sensitivity, AUC scores, and Youden index across sites. The ability of practitioners across sites to detect and characterize congenital anomalies using ultrasound varied according to body parts, with the circulatory and soft tissue anomalies presenting more diagnostic difficulties.

Technological innovations in medical imaging have transformed radiological diagnosis in the past two decades [31]. However, the expertise of the image interpreter is a key determinant of diagnostic efficacy in radiology [32,33]. The quality of the ultrasound systems and expertise of assessors involved in the morphological survey varied across sites and may have contributed to the variation in diagnostic performance. Two of the centers recorded considerably low sensitivity (58% and 62%), performed very poor in the detection of circulatory and facial anomalies, and involved sonographers with < 10 years of experience. These two centers use two-dimensional (2D) ultrasound systems (Mindray DP-3300 and Mindray DP-110 Plus) with no color Doppler function, and these may have contributed to the low detection rate. Previous studies have also shown that circulatory anomalies present diagnostic difficulty with accuracy scores as low as 50% [15,16,17]. One of these two centers demonstrated a PPV lower than 70%. Although the PPV depends on disease prevalence [34], abnormality prevalence was comparable across sites. Therefore, the low PPV recorded in this center is most likely due to intrinsic personnel or equipment limitations rather than low abnormality prevalence. Consistent with our study, previous works have reported wide variations in sensitivity and specificity for the detection of congenital anomalies, with these metrics varying according to the anatomical location of such anomalies. The highest diagnostic efficacy was observed for central nervous system (CNS) malformations where the sensitivity of ultrasound is about 90%. Studies reporting average ultrasound performance for all anatomical regions report sensitivity ranging from 71–99%, with specificity ranging from 45–99.9% [15,16,17]. Our results are well within the range reported in the literature [15,16,17], and the percentage misses are within that reported in radiology [35,36]. Inter-reader variability is a common occurrence in radiology [37,38], thus the variability in the current work is unsurprising.

Likelihood ratios provide a better understanding of the value of performing a diagnostic investigation [28]. We assessed the likelihood ratios to assess the usefulness of performing sonographic assessment of fetal morphology in our setting. The results showed high positive and extremely low negative likelihood ratios across facilities. Although the post-test probabilities observed indicate that fetal morphology ultrasound survey of congenital malformation is valuable, it should be remembered that disease prevalence affects the likelihood ratios and future probabilities [39]. There are not many well-powered population studies to provide a true understanding of the prevalence of fetal malformations in the Nigerian population, however, smaller studies have shown variability in geographic distribution of fetal anomalies, ranging from 0.4–2.7% [24,40,41]. Only one outlier reported a prevalence of 20.7 defects per 1000 populations [42]. Whilst the abnormality prevalence in our sample is consistent with a majority of these studies, data presented in the current work are from five facilities and prevalence should not be inferred to the Nigerian population. Additionally, the likelihood ratios and post-test probability reported should not be translated as the likelihood of a congenital malformation in the population, rather it should be interpreted as evidence of the value of ultrasound scan for fetal anomalies in the population. Alternatively, it should be interpreted as the usefulness of ultrasound for fetal morphology assessment taking into account the expertise available and future probabilities of anomalies. The Youden index, which accounts for all possible predictions [27], further supports the importance of fetal morphology assessment across the site surveyed.

Whilst the overall results demonstrate that fetal morphology assessment using ultrasound is valuable in low-economic settings, the low sensitivity and AUC values as well as inter-practice variation in diagnostic efficacy recorded suggest the need for interventions to improve diagnostic performance across sites, particularly for circulatory, soft tissue, and subtle abdominal wall anomalies, which accounted for a larger number of misses. Training on fetal morphology assessment, continuous professional development, and technologies that improve the detection of difficult anomalies may be important strategies to consider. A limitation of our work is that, we relied on postnatal clinical and surgical assessments to assess outcomes of livebirths. It is possible that subtle or minor internal anomalies may have been missed clinically and were not accounted for in our analysis. Nonetheless, our work does provide baseline data on the diagnostic efficacy of ultrasound personnel in fetal morphology evaluation and the value of such assessment in a resource-poor, non-screening setting. The outcome of the work can be used to develop models for training, monitoring and sustaining the accuracy of fetal morphology assessment in low-economic settings.

## 5. Conclusions

Fetal morphology assessment is valuable in poor economic settings, however, there is considerable variation in the diagnostic efficacy of facilities undertaking such assessment and a need to improve sensitivity for circulatory, facial, and soft tissue anomalies.

## Figures and Tables

**Figure 1 diagnostics-09-00109-f001:**
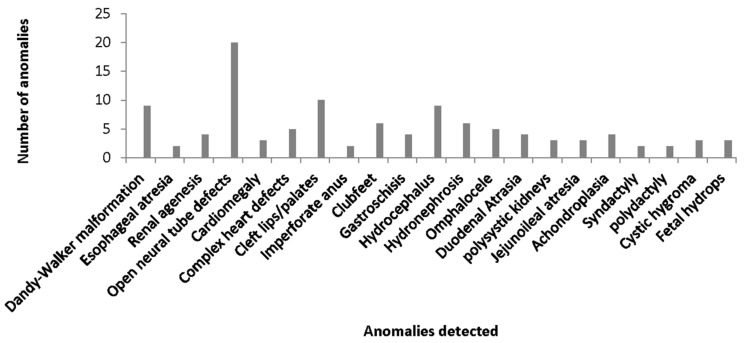
Distribution of anomalies detected.

**Figure 2 diagnostics-09-00109-f002:**
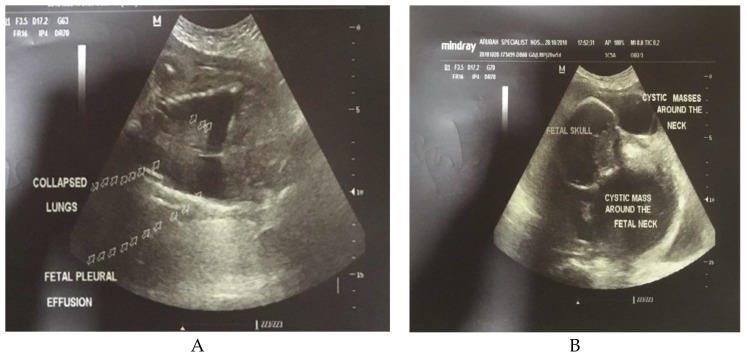
Ultrasound features of fetal hydrops (**A**) and Cystic hygroma (**B**).

**Figure 3 diagnostics-09-00109-f003:**
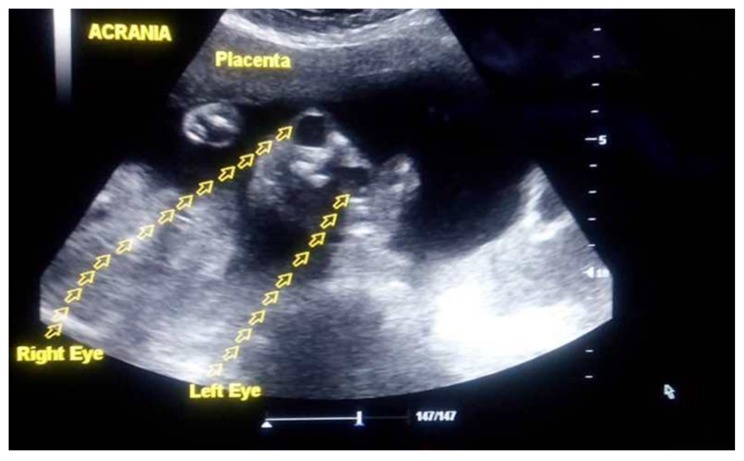
Ultrasound image of anencephaly.

**Table 1 diagnostics-09-00109-t001:** Distribution of anomalies according to body systems.

Body System Defects	Defects Detected on Ultrasound	Postnatal Diagnosis	Correctly Detected Using Ultrasound (%)
Nervous system	35 (41.7%)	38 (34.9%)	92.1
Genitourinary	10 (11.9%)	13 (11.9%)	76.9
Musculoskeletal	9 (10.7%)	12 (11.0%)	75.0
Digestive	10 (11.9%)	14 (12.8%)	71.5
Circulatory	4 (4.8%)	8 (7.3%)	50.0
Abdominal wall	6 (7.1%)	9 (8.3%)	66.7
Facial	7 (8.3%)	10 (9.2%)	70.0
Soft tissue	3 (3.6%)	5 (4.6%)	60
Total	84 (100%)	109	Mean: 70.3

**Table 2 diagnostics-09-00109-t002:** Average diagnostic efficacy across facilities.

Performance Metrics	Value	95% Confidence Interval
Sensitivity	77.1	68.0–84.6
Specificity	99.5	99.3–99.7
Positive Likelihood ratio	254.4	107.7–221.4
Negative Likelihood ratio	0.23	0.16–0.33
Abnormality prevalence	1.67 *	1.37–2.01
Positive predictive value	72.41	64.7–79.0
Negative predictive value	99.6	99.5–99.7
Area Under Curve	88.3	83.7–92.2
Youden index	77.1	

* = The prevalence of anomalies is not representative of the true population prevalence. Therefore, these values should not be inferred to the population.

**Table 3 diagnostics-09-00109-t003:** Summary of results across the five centers at 95% Confidence Interval.

Centers	Sensitivity	Specificity	PLR	NLR	Prevalence	PPV	NPV	AUC	J
C1	58(28,85)	99.7(99,100)	209.7(49,907)	0.42(0.21,0.82)	* 1.6(0.85,2.9)	77.7(45,94)	99.3(98.7,99.6)	78.9(63.5,92.5)	57.7
C2	73(45,92)	99.6(98.7,99.9)	168.7(52,543)	0.27(0.12,0.62)	* 2.1(1.2,3.5)	78.6(53,93)	99.4(98.7,99.8)	86.3(71.5,95.6)	72.6
C3	78(56,93)	99.5(99,100)	160.6(65,395)	0.22(0.1,0.47)	* 2.2(1.4,3.3)	78.3(59,90)	99.5(98.9,99.8)	88.8(77.5,96.5)	77.5
C4	62(47,76)	99.5(99,99.7)	121(69,215)	0.38(0.26,0.55)	* 1.6(1.2,2.2)	66.7(53,78)	99.8 (99,99.6)	80.8(73.4,75.2)	61.5

Numbers in brackets represent 95% confidence intervals. PLR: positive likelihood ratio; NLR: Negative likelihood ratio; PPV: positive predictive value; NPV: negative predictive value; AUC: Area Under the Curve. * = The prevalence of anomalies may not be representative of the true population prevalence. Therefore, these values should not be inferred to the population; J: Youden index.

## Data Availability

Data cannot be shared publicly due to consent and ethics approval restrictions on data deposition in public databases. Data are available from the Senior investigator (EUE) via his Shared folder for researchers who meet the criteria for access to confidential data upon ethics amendment.

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
