# Peer review of "Value and Diagnostic Efficacy of Fetal Morphology Assessment Using Ultrasound in a Poor-Resource Setting"

_diagnostics, 2019, doi:10.3390/diagnostics9030109_

Round 1

Reviewer 1 Report

[General comments]

This manuscript assessed the value and efficacy of fetal ultrasound in a Nigerian setting. The findings may have important clinical value for fetal ultrasound scanning in resource-limited countries or regions.

[Specific comments]

Page 1, Abstract. “and its value and efficacy fetal morphology” => “and its value and efficacy in fetal morphology” Page 1, Abstract. “, however” => “; however,” Page 2. “, however” => “; however,” Page 2. “for early detection of due to” Please add something after “of” Page 3. “Shenzhen Mindray Bio-Medical Electronics Co., Ltd” => “Shenzhen Mindray Bio-Medical Electronics Co., Ltd., Shenzhen, Guangdong, China” Page 5. “are is shown” => “is shown” Page 6. “in all five centres”: Please describe the five centres in Materials and Methods. Page 8, Lines 2-4. Quotations were used for this sentence. Does this mean that the sentence was cited from some literature? If yes, please provide the reference. Page 8, Line 17. “, however,” => “; however,” Page 8, Line 19. “varied according body parts”: Do you mean “varied according to body parts”? Page 8, Line 28. The two centres did not use color Doppler imaging. Does this mean the other three centres use color Doppler imaging? Is any else ultrasound imaging mode used? Is it possible to add some of typical images such as color Doppler in the manuscript? Page 8, Line 33. “due intrinsic personnel” => “due to intrinsic personnel” Page 8, Line 41. “Inter-reader’” => “Inter-reader’s” Page 8, Line 50. “, however” => “; however,” Page 9, Line 63. “however” should be deleted. Page 9, Line 76. “, however” => “; however,” Page 9, Line 90. “All the declare” => “All the authors declare”

Author Response

Response to Reviewer 1 Comments

Dear Reviewer, thank you for your suggestions to our manuscript. We have made the changes suggested.

Point 1: Page 1, Abstract. “and its value and efficacy fetal morphology” => “and its value and efficacy in fetal morphology”

Response 1: Thank you for spotting this. We have now changed the phrase “and its value and efficacy fetal morphology” to read “and its value and efficacy in fetal morphology”

Point 2: Page 1, Abstract. “, however” => “; however,”

Response 2: “, however” has been changed to “; however,”

Point 3: Page 2. “, however” => “; however,”

Response 3: Page 2. “, however” has been changed to “; however,”

Point 4: Page 2. “for early detection of due to” Please add something after “of”

Response 4: Thank you, this has now been revised. It reads:

“Ultrasound is the frontline modality for early detection of congenital anomalies due to its high sensitivity, non-ionizing nature, cost-effectiveness and availability [14]”

Point 5: Page 3. “Shenzhen Mindray Bio-Medical Electronics Co., Ltd” => “Shenzhen Mindray Bio-Medical Electronics Co., Ltd., Shenzhen, Guangdong, China”

Response 5: “Shenzhen Mindray Bio-Medical Electronics Co., Ltd” has been changed to “Shenzhen Mindray Bio-Medical Electronics Co., Ltd., Shenzhen, Guangdong, China”

Point 6: Page 5. “are is shown” => “is shown”

Response 6: “are is shown” has been changed to “is shown”

Point 7: Page 6. “in all five centres”: Please describe the five centres in Materials and Methods.

Response 7: We have now described the five centres in the “Material and Methods” section. It reads:

“Two of these five centres are specialist hospitals and provide ultrasound services to their patients and clients referred from other health facilities. The other three centres are private diagnostic facilities providing diagnostic imaging and laboratory services to patients from public and private health facilities.” 

Point 8: Page 8, Lines 2-4. Quotations were used for this sentence. Does this mean that the sentence was cited from some literature? If yes, please provide the reference. 

Response 8: Thank you for pointing this out. This is not a direct quote. We have now removed the quotation marks and a reference provided to support the statement. It now reads: “The post-test probability represents the revised probability of detecting an anomaly after taking into account new gestations [23]”

Point 9: Page 8, Line 17. “, however,” => “; however,”

Response 9: “, however,” has been changed to “; however,”

Point 10: Page 8, Line 19. “varied according body parts”: Do you mean “varied according to body parts”?

Response 10: Page 8, Line 19. “varied according body parts” has been changed to “varied according to body parts”

Point 11: Page 8, Line 28. The two centres did not use color Doppler imaging. Does this mean the other three centres use color Doppler imaging? Is any else ultrasound imaging mode used? Is it possible to add some of typical images such as color Doppler in the manuscript?

Response 11: Thank you for your comment and suggestion. Yes, the other three centres use scanners with color Doppler facilities. However, we did not assess whether or not Doppler was used in assessing these fetuses. Since Doppler is helpful for vascular anomalies, we assumed that a lack of it the two facilities highlighted may have contributed to decreased diagnostic accuracy. Thus, the statement that “…and these may have contributed to the low detection rate.” was to suggest that there is a possibility that a lack of Doppler facilities in the other two centres may have negatively impacted diagnostic efficacy for vascular anomalies, but we are not certain that this is true since we did not assess whether or not other centres performed Doppler studies for all fetuses with cardiovascular anomalies.

We would have loved to include Doppler images. However, due to storage limitations, the facilities surveyed do not store images of patients, and images are often printed in greyscale. In addition, the rarity of these makes it difficult to obtain new images within the time given for the revision. This has made it difficult to include Doppler images in the manuscript.

Point 12: Page 8, Line 33. “due intrinsic personnel” => “due to intrinsic personnel”

Response 12: “due intrinsic personnel” has been changed to “due to intrinsic personnel”

Point 13: Page 8, Line 41. “Inter-reader’” => “Inter-reader’s”

Response 13: “Inter-reader’” has been changed to “Inter-reader”

Point 14: Page 8, Line 50. “, however” => “; however,”

Response 14: “, however” has been changed to “; however,”

Point 15: Page 9, Line 63. “however” should be deleted.

Response 15:  “however” has been deleted

Point 16: Page 9, Line 76. “,however” => “; however,”

Response 16: “, however” has been changed to “; however,”

Point 17: Page 9, Line 90. “All the declare” => “All the authors declare” 

Response 17: “All the declare” has been changed to “All the authors declare” 

Reviewer 2 Report

The aim of the manuscript was to evaluate the value and diagnostic efficacy of fetal morphology assessment with the use of ultrasound in poor economy settings. The authors surveyed 6250 fetuses using the ISUOG guidelines in the period of three years. Unfortunately, the study included only three variables regarding medical history, i.e.:

maternal age - with very broad range from 15 to 46 years gestational age - 20 +/- 2 weeks of gestation number pregnancies - an enigmatic piece of information: "Parity status ranged from single to multiparous, and 70% of these women had at least one child prior to the fetus surveyed in the current work".

One of the limitations of this study is the fact that authors did not include other aspects of medical history which are important for the risk of congenital defects, for example maternal chronic and gestational diseases,  treatment during the first and second trimesters, familial hereditary diseases, maternal bad habits.

Authors repeadetly emphasized that the prevalence of fetal anomalies was not representative of the true population prevalence. Did any previous studies determine the population prevalence of fetal malformations in Nigeria?

The sentence "Abnormality prevalence in the cohort surveyed varied from 1.6% (95%CI: 0.85–2.9) to 2.2% (95%CI:1.4–3.3)." (page 5) is difficult to understand (2.2% - ??).

Fetal morphology assessment was conducted by five radiologists with varied previous experience. How many ultrasound scans of fetuses at 20 weeks of gestation were performed by each operator before the study?

The interest of readers would prove higher if they could find some additional information regarding current obstetric care in Nigeria:

How many pregnant women (what percentage of them) have an opportunity to perform ultrasound of their fetuses in Nigeria? How many centers survey such ultrasound scans in Nigeria? Are there any programs to fund such tests for pregnant women? Are only radiologists performing ultrasound on pregnant women in Nigeria?

Author Response

Response to Reviewer 2 Comments

Dear Reviewer, Thank you for your time in reviewing our manuscript and for your comments. Please find below our responses and clarifications to your comments.

Point 1: The aim of the manuscript was to evaluate the value and diagnostic efficacy of fetal morphology assessment with the use of ultrasound in poor economy settings. The authors surveyed 6250 fetuses using the ISUOG guidelines in the period of three years. Unfortunately, the study included only three variables regarding medical history, i.e.: maternal age - with very broad range from 15 to 46 years gestational age - 20 +/- 2 weeks of gestation number pregnancies - an enigmatic piece of information: "Parity status ranged from single to multiparous, and 70% of these women had at least one child prior to the fetus surveyed in the current work".

One of the limitations of this study is the fact that authors did not include other aspects of medical history which are important for the risk of congenital defects, for example maternal chronic and gestational diseases, treatment during the first and second trimesters, familial hereditary diseases, maternal bad habits.

Response 1: Thank you for your observations. These observations are valid only if we were interested in assessing the prevalence of congenital anomalies or risk factors for congenital anomalies in the population. We have stated categorically and re-emphasized that this is not a prevalence or risk assessment study, rather we are interested in understanding whether or not facilities undertaking fetal morphology assessment can perform fetal morphology assessment using ultrasound, and if ultrasound imaging is a valuable tool for fetal anomaly survey in our region (diagnostic efficacy). It is well established that clinical history and patient age do not have any impact on diagnostic imaging accuracy.  Therefore, it is highly unlikely that the recruitment of women of maternal ages 15 to 46 years and the non-inclusion of “aspects of medical history which are important for the risk of congenital defects, will have any impact on the validity or generalizability of the findings of this study.

In summary, the information considered as a limitation of our study is relevant when undertaking risk assessment NOT for diagnostic accuracy and value of an imaging tool.

We have now revised the statements considered to be enigmatic to enhance clarity. They read:

“Morphological surveys of all fetuses were performed between the 18th to the 22nd week of pregnancy according to the ISUOG guideline [26]”

 “A total of 4,375 of the women surveyed had at least one child (range 1 – 4 children) prior to the current study”

Point 2: Authors repeatedly emphasized that the prevalence of fetal anomalies was not representative of the true population prevalence. Did any previous studies determine the population prevalence of fetal malformations in Nigeria?

Response 2: Thank you for your comment. As stated above, we were interested in the value and diagnostic efficacy of ultrasound in fetal morphology assessment, NOT the prevalence of fetal anomalies. Our emphasis that prevalence of fetal anomalies is not representative of the true population prevalence is because we do not want readers to infer population prevalence from our study. We had also provided the answer to your question in the discussion section. It reads:

“There are not many well powered population studies to provide a true understanding of the prevalence of fetal malformations in the Nigerian population; however smaller studies have shown variability in geographic distribution of fetal anomalies, ranging from 0.4–2.7% [18, 34, 35]. Only one outlier reported a prevalence of 20.7 defects per 1,000 populations [36].”

Point 3: The sentence "Abnormality prevalence in the cohort surveyed varied from 1.6% (95%CI: 0.85–2.9) to 2.2% (95%CI:1.4–3.3)." (page 5) is difficult to understand (2.2% - ??).

Response 3: Thank you for pointing this out. We surveyed five sites in different cities, and the range of prevalence reported represents prevalence across these sites. We have now revised this sentence to enhance clarity. It reads:

“Abnormality prevalence across the different sites surveyed varied from 1.6%; 95%CI: 0.85–2.9 (centre 1) to 2.2%; 95%CI: 1.4–3.3 (centre 3).”

Point 4: Fetal morphology assessment was conducted by five radiologists with varied previous experience. How many ultrasound scans of fetuses at 20 weeks of gestation were performed by each operator before the study?

Response 4: Thank you for your question. We have now added this information. It reads:

“Each of the assessors had performed a minimum of 1,000 ultrasound examinations of fetuses at 20 weeks of gestation per year within the last three years prior to the current study.”

Point 5: The interest of readers would prove higher if they could find some additional information regarding current obstetric care in Nigeria.

How many pregnant women (what percentage of them) have an opportunity to perform ultrasound of their fetuses in Nigeria? How many centers survey such ultrasound scans in Nigeria? Are there any programs to fund such tests for pregnant women?

Response 5: Thank you for your suggestions. There is currently no statistics on the number of facilities performing morphology ultrasound scans or the percentage of women that have an opportunity to perform ultrasound; however, it has been shown that women in suburban and urban regions have good attitude to obstetric ultrasound uptake. We have now added the information requested about current obstetric care in Nigeria. It reads:

“There is considerable variation in obstetric care in Nigeria due to differences in socio-cultural, economic, and geographical distribution of the population [18, 19]. Women in rural areas access antenatal services provided by community health centres, which often do not have the facilities and expertise to undertake detailed obstetric assessment and care[18, 19]. Access to health antenatal services is further challenged by the poor economic status and deep-rooted socio-cultural beliefs [19, 20]. With the Mother and Child Initiative of the General Electric Healthcare, where community nurses are trained to detect risk in pregnancy using ultrasound, obstetric care is expected to improve. Conversely, women in suburban and urban regions have access to public and private antenatal services, and have been shown to demonstrate good uptake of obstetric ultrasound assessment [21, 22]. Studies have also shown improvement obstetric outcomes for women in these regions [23]; however, progress has been slow, partly due to the absence of government-funded programs to improve obstetric care.”

Point 6: Are only radiologists performing ultrasound on pregnant women in Nigeria? 

Response 6: We had already provided this information in the introduction. It reads:

“Due to a paucity of radiologists, most ultrasound investigations are performed by sonographers and resident radiologists, and some of these personnel have limited training or experience in ultrasonography [18, 19]”.

Details of the personnel that performed the ultrasound examinations used for the current study are also shown in the methods section. It reads:

“Of the five facilities surveyed, four were operated by sonographers (two in one facility, and one each in the other three facilities) at different stages of practice and the other by senior resident radiologists (n=3)”

Round 2

Reviewer 2 Report

Dear Authors,

Thank you for attempting to address some of my concerns. 

Unfortunately, the aim of the manuscript, i.e. "understanding whether or not facilities undertaking fetal morphology assessment can perform fetal morphology assessment using ultrasound, and if ultrasound imaging is a valuable tool for fetal anomaly survey in the authors' region (diagnostic efficacy)" seems to be ... OBVIOUS.

My question (Number 6) was associated with the fact that during reading of your manuscript I was surprised that OBSTETRICIANS AND GYNECOLOGISTS are not included in the group of specialists who perform ultrasound of pregnancies in Nigeria.

References have not been described according to the instructions for authors.

If the limitations of this manuscript are not important for the Editor, I am able to accept the manuscript after references correcting.

Author Response

Reviewer 2

Point1: My question (Number 6) was associated with the fact that during reading of your manuscript I was surprised that OBSTETRICIANS AND GYNECOLOGISTS are not included in the group of specialists who perform ultrasound of pregnancies in Nigeria.

Response 1:  Thank you for reviewing our manuscript and providing important suggestions for improvement. We now understand the reason for this question. OBSTETRICIANS AND GYNECOLOGISTS are not involved in obstetrics and gynecology ultrasound in Nigeria.

Point 2: References have not been described according to the instructions for authors.

Response 2: Thank you. We have now made corrections to the reference list to conform to the MDPI reference style as shown in the example below:

“Egbe, A.; Lee, S.; Ho, D.; Uppu, S.; Srivastava, S. Racial/ethnic differences in the birth prevalence of congenital anomalies in the United States. J Perinat Med, 2015, 43, 111-7.”
